# Screening for the Key Proteins Associated with Rete Testis Invasion in Clinical Stage I Seminoma via Label-Free Quantitative Mass Spectrometry

**DOI:** 10.3390/cancers13215573

**Published:** 2021-11-08

**Authors:** Lucia Borszéková Pulzová, Jan Roška, Michal Kalman, Ján Kliment, Pavol Slávik, Božena Smolková, Eduard Goffa, Dana Jurkovičová, Ľudovít Kulcsár, Katarína Lešková, Peter Bujdák, Michal Mego, Mangesh R. Bhide, Lukáš Plank, Miroslav Chovanec

**Affiliations:** 1Biomedical Research Center, Department of Genetics, Cancer Research Institute, Slovak Academy of Sciences, Dúbravská cesta 9, 845 05 Bratislava, Slovakia; lucia.pulzova@savba.sk (L.B.P.); jan.roska@savba.sk (J.R.); eduard.goffa@savba.sk (E.G.); dana.jurkovicova@savba.sk (D.J.); ludovit.kulcsar@savba.sk (Ľ.K.); michal.mego@nou.sk (M.M.); 2Department of Pathological Anatomy, Jessenius Faculty of Medicine and University Hospital in Martin, Comenius University, Malá Hora 4A, 036 01 Martin, Slovakia; kalman@unm.sk (M.K.); slavikpavol@centrum.sk (P.S.); leskova45@uniba.sk (K.L.); lukas.plank@uniba.sk (L.P.); 3Clinic of Urology, Jessenius Faculty of Medicine and University Hospital in Martin, Comenius University, Malá Hora 4A, 036 01 Martin, Slovakia; jan.kliment1@uniba.sk; 4Biomedical Research Center, Department of Molecular Oncology, Cancer Research Institute, Slovak Academy of Sciences, Dúbravská cesta 9, 845 05 Bratislava, Slovakia; bozena.smolkova@savba.sk; 5Department of Urology, Faculty of Medicine, Comenius University, 813 72 Bratislava, Slovakia; peter.bujdak@kr.unb.sk; 62nd Department of Oncology, Faculty of Medicine, Comenius University and National Cancer Institute, Klenová 1, 833 10 Bratislava, Slovakia; 7Department of Microbiology and Immunology, University of Veterinary Medicine, Komenského 73, 041 81 Košice, Slovakia; mangesh.bhide@uvlf.sk; 8Institute of Neuroimmunology, Slovak Academy of Sciences, Dúbravská cesta 9, 845 05 Bratislava, Slovakia

**Keywords:** testicular germ cell tumours, clinical stage I seminoma, rete testis invasion, proteomics, mesenchymal type proteins

## Abstract

**Simple Summary:**

For clinical stage I (CS I) seminoma patients, management through the risk-adapted strategy with adjuvant carboplatin-based chemotherapy in the presence of risk factors and surveillance in the absence of these factors is the preferred option. In such management, rete testis invasion (RTI) represents a prognostic factor, as its absence, together with a tumour diameter ≤4 cm is associated with a very low relapse risk. To be able to routinely manage CS I seminoma patients through a risk-adapted strategy, reliable biomarkers stratifying the risk of relapse for CS I seminoma patients are urgently required. However, no such biomarker has yet entered routine use in clinical decision-making or clinical guidelines. The lack of consistent prognostic biomarkers for CS I seminoma patients prompted us to compare the proteomic profiles of RTI-positive and -negative CS I seminomas to reveal the molecular mechanism(s) and, in particular, the corresponding biomarkers of RTI invasion.

**Abstract:**

Rete testis invasion (RTI) is an unfavourable prognostic factor for the risk of relapse in clinical stage I (CS I) seminoma patients. Notably, no evidence of difference in the proteome of RTI-positive vs. -negative CS I seminomas has been reported yet. Here, a quantitative proteomic approach was used to investigate RTI-associated proteins. 64 proteins were differentially expressed in RTI-positive compared to -negative CS I seminomas. Of them, 14-3-3γ, ezrin, filamin A, Parkinsonism-associated deglycase 7 (PARK7), vimentin and vinculin, were validated in CS I seminoma patient cohort. As shown by multivariate analysis controlling for clinical confounders, PARK7 and filamin A expression lowered the risk of RTI, while 14-3-3γ expression increased it. Therefore, we suggest that in real clinical biopsy specimens, the expression level of these proteins may reflect prognosis in CS I seminoma patients.

## 1. Introduction

Clinical stage I (CS I) seminoma patients have a very good prognosis, with an overall survival (OS) rate of 98%. These patients are characterized by disease confined to the testicle, postorchiectomy normalized tumour markers, negative imaging of the chest, abdomen, and pelvis, and a normal physical examination. For these patients, radical inguinal orchiectomy followed by close surveillance is standard management, under which, however, about 15–20% of patients develop tumour recurrence in follow-up (reviewed in [1,2]). Adjuvant therapy managed by chemotherapy or radiation therapy represents an option to reduce the risk of relapse in CS I seminoma patients. Indeed, both irradiation of the equilateral retroperitoneal lymphatic tissue and intravenous administration of carboplatin (1 or 2 cycles) lower relapse risk to less than 5% [3,4,5].

There is a possibility of late negative effects, such as second malignant neoplasm, cardiovascular disease, neurotoxicity, nephrotoxicity, pulmonary toxicity, hypogonadism, decreased fertility and psychosocial problems [6], when adjuvant therapy for CS I seminoma patients is applied [7]. Therefore, consistent prognostic factors for the prediction of relapse, and thus for guiding the management of these patients after orchiectomy—particularly for counselling them about adjuvant treatment—are urgently required. This need is further supported by the fact that the second relapse rate is higher in patients relapsing after adjuvant treatment than in those relapsing under surveillance [8,9]. After the very first prognostic factors, histopathological characteristics of the primary tumour specimen, the primary tumour size and rete testis invasion (RTI) were associated with tumour recurrence [10,11] and recommended for use in clinical practice to guide decision making on adjuvant treatment [12]. However, when systematically reviewed [1,2], their prognostic power has been found to be too weak to advocate their routine clinical use. In line with this, some authorities do not clearly recommend the primary tumour size and RTI as prognostic factors for the decision on adjuvant treatment for CS I seminoma patients because of limited and inconclusive evidence [13,14]. Several other prognostic factors have also been examined, such as age at diagnosis, preoperative tumour marker levels, testicular vascular invasion, tumour necrosis, albuginea penetration, epididymis invasion, base of cord invasion, and vascular invasion of cord, but all of them have been found to be associated too weakly with the relapse risk for CS I seminoma patients under surveillance [15,16]. Therefore, there are no obvious prognostic factors reliably stratifying CS I seminoma patients for risk of relapse yet.

Predictive markers of metastatic disease are essential in customizing clinical management for cancer patients. Of primary importance in prognosis of cancer patients is the sequence of events leading to the development of tumour invasion of the surrounding tissues and metastasis. The metastatic cascade can broadly be separated into three main processes: invasion of surrounding tissue, intravasation and extravasation. This complex process employs the transformation of adherent epithelial cells to motile mesenchymal cells (epithelial–mesenchymal transition; EMT). The course of EMT involves an alteration of characteristic epithelial cell morphology and gene expression patterns, resulting in a mesenchymal phenotype and acquisition of migratory and invasive properties. In tumour tissue, cancer cells most often undergo an incomplete EMT, where both epithelial and mesenchymal markers can be detected in the same cancer cell at the same time. The complexity of EMT and variability in the stage of EMT leads to a wide range of different profiles of EMT markers. In a local context, a critical step in cancer progression is surrounding tissue invasion (stromal compartment), where epithelial cells can be exposed to the stromal extracellular matrix (ECM), which is distinct from the ECM within the normal epithelial compartment (reviewed in [17]). In testicular germ cell tumours (TGCTs), RTI has been identified as a predominant pathway of extratesticular extension associated with metastatic progression, even though its significance still remains rather controversial and is awaiting conclusive confirmation [18,19].

The application of mass-based technology to whole proteome analysis is a widely used approach in the comprehensive detection and characterization of proteins. It provides information on protein expression and overcomes the limitations of immunohybridization and restriction of range of the measured protein levels found in microarray technologies. Generation of publicly available large-scale datasets, such as The Cancer Genome Atlas (TCGA), provides comprehensive catalogues of multiple data types performed on the same set of samples. Various groups have identified large gene signatures that are prognostic for outcomes or chemotherapeutic response in profiled human cancer samples through the TCGA dataset [20,21,22].

The present study was aimed at comparing the proteomic profile of RTI-positive and -negative CS I seminoma patient tissues in order to gain insights into the molecular mechanism(s) underlying invasive phenotypes. The proteins with altered expression were subjected to gene ontology (GO) enrichment analysis and compared with published proteomic and transcriptomic data sets. Selected differentially expressed proteins were clinically validated using CS I seminoma tumour specimens with known RTI status. To assess the significance of these proteins for the process of metastasis, expression of the corresponding genes in primary tumour- and metastasis-derived TGCT cell lines was also examined.

## 2. Materials and Methods

### 2.1. Research Ethics

All subjects gave their informed consent for inclusion before they participated in the study. The study was conducted in accordance with the Declaration of Helsinki, and the protocol was approved by the Ethics Committee of Jessenius Faculty of Medicine in Martin, Comenius University in Bratislava (Protocol Nr. EK 52/2020).

### 2.2. Tissue Collection and Sample Preparation for LC–MS/MS

For LC–MS/MS analysis, samples from 6 tumour tissues (3 with RTI-positive and 3 with -negative CS I seminoma), who underwent curative orchiectomy in the Department of Urology, Slovak Medical University, were used. Patients did not receive treatments other than surgical therapy. During the excision surgery, 2 g of fresh cancer tissue was obtained from each patient. Discrimination between cancer tissues and the adjacent tissues was made by pathological examination. Tissue samples were thoroughly cleaned from blood, fat and connective tissues under the inverted microscope. Subsequently, the tissues were washed with phosphate-buffered saline (PBS) under sterile conditions and immediately frozen down in liquid nitrogen and stored in liquid nitrogen until used.

Tissue samples were cut into small pieces (~1–2 mm) and the subcellular fractions from tissue samples were prepared using a ProteoJET Membrane Protein Extraction Kit (ThermoFisher Scientific, Rockford, IL, USA) according to the manufacturer’s instructions. In brief, frozen tissues were ground in liquid nitrogen using a pestle and resuspended in 2 mL of ice-cold cell permeabilization buffer. The mixture was incubated for 10 min at 4 °C with shaking. Permeabilized cells were centrifuged at 16,000× *g* for 15 min at 4 °C, and the cytoplasmic fraction (supernatant) was separated from the membrane fraction (pellet). Cytoplasmic fractions were further treated with ProteoBlock ™ Protease Inhibitor Cocktail (ThermoFisher Scientific, Rockford, IL, USA). Protein concentration of clarified cell lysates was determined using a bicinchoninic acid protein assay kit (ThermoFisher Scientific, Rockford, IL, USA). Finally, a protein pool was prepared by using 200 µg of total protein extracted from each CS I seminoma sample.

### 2.3. Protein Digestion and LC–MS/MS

100 µg of cytoplasmic fraction was digested with Trypsin gold (Promega, Madison, WI, USA) and labelled with tandem mass tag reagents (ThermoFisher Scientific, Rockford, IL, USA) according to the manufacturer’s instructions. After labelling, samples were clubbed to make two groups: RTI-positive and -negative CS I seminomas. Samples were combined in equal amounts. One forth volume of the combined sample was pre-fractionated using strong cation exchange (Ettan LC, GE Healthcare). Fractionated samples were then loaded in nanoLC–MS/MS. For mass spectrometry, the peptide precursor mass tolerance was set at 10 ppm, and MS/MS tolerance at 0.8 Da. Search criteria included oxidation of methionine (+15.9949) as a variable modification, carbamidomethylation of cysteine (+57.0214) and the addition of isobaric mass tags (+229.163) to peptide N-termini and lysine as fixed modifications. In searches, a maximum of 1 missed cleavage was allowed and the search was performed with full tryptic digestion. The reverse database search option was enabled, and all peptide data were filtered to satisfy a false discovery rate of 5%. Quantitation was performed using a peak integration window tolerance of 0.0075 Da with the integration method set as the most confident centroid. Protein ratios represent the median of the raw measured peptide ratios for each protein. The raw data files were processed and quantified using Proteome Discoverer software v1.2 (Thermo Scientific) and searched against the UniProt/SwissProt Human database using the SEQUEST algorithm. Each protein included in our study was identified from at least 2 peptides with high/medium confidence. Proteins recorded as uncharacterized by the software were returned with a gene ID.

### 2.4. GO and Pathway Analysis

The GO-annotated proteome was derived from the UniProt-GOA database. A protein fold change greater than 1.2 or less than 0.8 was considered as indicating a differentially abundant protein.

GO analysis using Database for Annotation, Visualization and Integrated Discovery (DAVID) [23] web tool was performed for the differentially expressed proteins between the RTI-positive and -negative CS I seminoma, to show the major biological and molecular factors and mechanisms that were affected by the invasive phenotype. For each CS I seminoma sample, the proteins were separated into lists of proteins which were up-regulated and down-regulated. Each list was uploaded into the DAVID 6.7 web tool for separate analysis to generate the list of biological and molecular functions impacted by changes in protein expression. First, the Blast2GO was used to download the annotated human protein data from the NCBI database and functional classification was performed for every annotated protein.

Additionally, we used The Human Protein Atlas (http://www.proteinatlas.org (originally accessed on 13 June 2019; revised and updated on 19 August 2021) to perform validation of expression, and prognosis of the candidate proteins. Furthermore, the expression levels of these proteins in testis tumours available at The Human Protein Atlas were compared to shortlist the targets for validation.

### 2.5. Patient Samples and Cell Lines

All patients enrolled into this study underwent orchiectomy in one of several hospitals across Slovakia (in the period between the years 2008–2019) and were diagnosed with CS I seminoma. Patient samples were represented by 10% buffered formalin-fixed, paraffin-embedded orchiectomy specimens. The samples were identified in the biopsy register and collected from the biopsy archive of the department.

Primary tumour- and metastasis-derived TGCT cell lines (2102EP and 1777NRpmet, respectively) were kindly provided Dr. Thomas Mueller (University Clinic for Internal Medicine IV, Hematology/Oncology, Medical Faculty of Martin Luther University Halle-Wittenberg, Halle, Germany). Histologically, 210EP is an embryonal carcinoma and 1777NRpmet is a differentiated embryonal carcinoma with immature teratoma [24,25]. Both cell lines were grown in RPMI-1640 medium supplemented with 10% fetal bovine serum, penicillin (100 U/mL) and streptomycin (10 μl/mL). Cell lines were cultivated at 37 °C in 5% CO_2_ atmosphere [26].

### 2.6. Immunohistochemistry (IHC)

Formalin-fixed paraffin-embedded 4 µm thick tissue sections were used for IHC detection of the selected proteins with the following monoclonal antibodies (all produced by Santa Cruz Biotechnology, Inc., USA): 14-3-3γ (clone D-6 sc-398423), 14-3-3β (clone A-6 sc-25276), PARK7 (clone D-4 sc-55572), ezrin (clone 3C12 sc-58758), vinculin (clone 7F9 sc-73614), vimentin (clone 5G3F10 sc-66002), filamin A (clone E-3 sc-17749) and caldesmon (clone A-2 sc-271222). Antibodies were used according to the manufacturer’s recommendations. Sections were revitalized in the automated pre-treatment link (DakoDenmark A/2, Glostrup Denmark) in the HpH (pH = 9) or LpH (pH = 6) solution under a temperature of 97 °C for 20 min, followed by IHC reactions run in an Autostainer Link 48 (DakoDenmark A/2, Glostrup Denmark). For visualization, a detection EnVision kit and DAB were used, followed by a final contrast hematoxylin staining.

Protein expression evaluated by IHC was monitored in both the tumour and the intratumoural/stromal immune mononuclear cells (TC and IC, respectively). The positive expression was recorded as nuclear (N), membranous (M), paranuclear (P) and cytoplasmic (C). The expression intensity was graded in 3 grades: tier 0 (absent expression), 1+ (weak expression) and 2+ (strong expression). To evaluate IHC data, protein expression was further individually categorized as follows: 14-3-3γ (TC positivity)—only category 0 (no expression) and 2 (diffuse expression) was scored, as category 1 (focal expression) was not observed; ezrin (IC positivity)—only category 0 (no expression) and 2 (diffuse expression) was scored, as category 1 (focal expression) was not observed; filamin A (TC positivity)—categorized as 0 (low expression; sum of grade 0 and 1+) and 1 (high expression; grade 2+); PARK7 (TC positivity)—expression categories 0 (grade 0), 1 (grade 1+) and 2 (grade 2+); vimentin (TC positivity)—only category 0 (no expression) and 1 (focal expression) was scored, as category 2 (diffuse expression) was not observed; and vinculin (IC positivity)—0 (low expression) and 1 (high expression).

The evaluation was performed by the methods of 3 independent pathologists (M.K., P.S. and L.P.), in the case of disagreement the cases were discussed while using multihead microscope.

### 2.7. mRNA Expression Analysis in TGCT Cell Lines

To examine differences in mRNA expression, TRI Reagent solution (Life Technologies, Carlsbad, CA, USA) was used for total RNA extraction. Isolated RNA was quantified using a MaestroNano Spectrophotometer (Applied Biological Materials Inc., Richmond, British Columbia, CA). Relative expression of the *YWHAB*, *YWHAG, CALD1*, *EZRI*, *FLNA*, *PARK7*, *VINC* and *VIME* genes was evaluated by RT-qPCR using a First-strand cDNA Synthesis System (Central European Biosystems) for reverse transcription. For cDNA synthesis, 1.5 μg of total RNA, 2 μL of 10× MuLV buffer, 1 μM of p(dN)6 primer, 0.1 mM of dNTP mix and 100 units of MuLV reverse transcriptase were incubated at 42 °C for 1 h followed by enzyme inactivation at 70 °C for 5 min. Real-time PCR detection and quantification of expression of the above-mentioned genes, as well as of the *PGK1* (phosphoglycerate kinase 1) reference gene was performed using SYBR Premix Ex Taq II (Tli RNaseH Plus), ROX plus (Takara) and primers listed in Appendix A. Acquired Ct (cycle threshold—defined as the number of cycles required for the fluorescent signal to cross the threshold) values were normalized against the *PGK1* reference gene, which showed stable expression across both TGCT cell lines. The mean ± SD values of Ct for the *PGK1* gene were 21.5 ± 0.059 and 21.54 ± 0.497 for 2102EP (primary tumour-derived) and 1777NRpmet (metastasis-derived) TGCT cell lines, respectively.

### 2.8. Statistical Analysis

The patient clinical characteristics were categorized as frequency (percentage). Pearson chi-square or Fisher exact tests were used to examine the association between RTI and clinical characteristics or protein expression. Logistic regression was applied to identify variables associated with RTI. Each model included maximum tumour diameter, TNM staging, age at diagnosis, and proteins found significant by univariate analysis. A backward model selection was conducted, and the final fitted model is presented. The statistical analyses were performed by the software IBM SPSS statistics, version 25.0 (IBM Corp. Armonk, NY, USA). The data are shown as the mean ± SD and were analysed by Student’s *t*-test. *p* values < 0.05 were considered to be significantly different.

Statistical analysis of mRNA and protein expression in TGCT cell lines was performed using SigmaPlot 12.5. Normality of the data distribution was tested by the Shapiro–Wilk test. In the case of mRNA expression, relative fold changes were calculated using 2-ΔΔCt method, where ΔΔCt = ΔCt (metastasis-derived TGCT cell line)2014ΔCt (primary tumour-derived TGCT cell line). Data are presented as mean (1777NRpmet vs. 2102EP), with error bars representing upper and lower limits of expression (2^−ΔΔCt ± SD^) of three technical and three biological replicates. For analysis of significance of fold changes in mRNA expression between the two TGCT cell lines, ΔCt values were used. If normally distributed, the mRNA expression data were tested by two-tailed *t*-test. For non-normally distributed data, Mann–Whitney U tests were used. For all analyses, *p* value < 0.05 was considered as statistically significant (* *p* < 0.05; ** *p* < 0.01; *** *p* < 0.001).

In TCGA analysis, OS survival was estimated by the Kaplan–Meier method and compared using the log-rank test.

## 3. Results

### 3.1. RTI Promotes Protein Expression Change

Six CS I seminoma patient samples (three RTI-positive and three -negative) were used in the proteomic analysis aimed at revealing factors that are differentially expressed upon RTI. The median patient age at surgery was 35.5 years (ranging from 28 to 50), 29 (ranging from 28 to 39) for RTI-positive and 37 (ranging from 34 to 50) for -negative patients. The average excised tumour size was 41 mm (ranging from 17 to 70), 37 (ranging from 17 to 70) for RTI-positive and 45 (ranging from 20 to 65) for -negative patients. Cytoplasmic protein fractioning and well-established label-free liquid chromatography/tandem mass spectrometry (LC–MS/MS) proteomic workflows were used to compare the overall protein expression profiles in RTI-positive with those in -negative CSI seminoma patients, and hence to identify proteins that were differentially expressed between the two groups. Proteins were considered only if identified by more than 1 peptide (the number of peptides for each protein used for quantification ranged from 1 to 19). In total, 64 proteins were found to be differentially expressed in RTI-positive vs. -negative CS I seminoma, where expression changes higher than 20% were taken as a cut-off criterion, with 44 and 20 proteins being up- and down-regulated, respectively (Appendix A).

### 3.2. GO Annotation and Functional Classification

Differentially expressed proteins were subjected to the DAVID web tool for the GO enrichment analysis. Analysis of differentially expressed proteins in RTI-positive vs. -negative CS I seminoma showed that proteins were enriched in certain molecular functions, biological processes and cellular components (Figure 1). According to molecular functions, these proteins were enriched in poly(A) RNA binding, and in cadherin and protein binding involved in cell–cell adhesion. Biological process classification suggested that cell–cell adhesion, and cell and biological adhesion were the dominant processes enriched in RTI-positive CS I seminoma. Based on the cellular compartments, differentially expressed proteins were mainly components of the extracellular exosome, vesicle and organelle (Figure 1). Among the differentially expressed proteins involved in the most significantly enriched GO terms category molecular function, 14-3-3β, 14-3-3γ, caldesmon, ezrin, filamin A, PARK7, vimentin and vinculin were subjected for further examination to gain a better understanding of the mechanisms that are the basis of the RTI-positive phenotype.

### 3.3. Database Search

The Human Protein Atlas database was queried to compare our results with the known expression levels associated with cancer. Expression of all studied proteins was reported for all cancer types deposited in TCGA and its prognostic value in cancer tissues is summarized in Table 1. In testicular cancer, expression of these proteins has not been associated with prognosis yet.

### 3.4. Validation of Mass Spectrometry Data by IHC

While 14-3-3γ, ezrin, filamin A, PARK7, vinculin and vimentin could be analysed in cohort of CS I seminoma patients using IHC to validate their clinical relevance, IHC evaluation in the case of caldesmon and 14-3-3β failed (data were non-homogenous and suboptimal). A total of 74 patients were analysed—37 with RTI and 37 without RTI (Table 2). The median patient age at surgery was 40.2 years (ranging from 20.2 to 61.1 years), 40.3 (ranging from 23.7 to 61.1) for RTI-positive and 38.6 (ranging from 20.2 to 55.7) for -negative patients. All patients had a good prognosis according to International Germ Cell Cancer Collaborative Group (IGCCCG) criteria. After orchiectomy, 32 (43.2%) patients were treated with carboplatin, 27 (36.5%) patients underwent radiation therapy and 12 (16.2%) patients were managed by surveillance strategy (i.e., absence of any adjuvant treatment). Two patients refused any treatment modality after surgery and in the case of one (4.1% for sum of no, or no evidence of, treatment modality) patient, information on treatment is lacking.

For statistical analysis of IHC data, protein expressions were categorized (Table 3; for further details on protein expression categorization, see Materials and Methods). Individual protein expressions were then compared between the RTI-positive and -negative CSI seminoma patients. High protein expression was more frequent in RTI-positive than in -negative patients for 14-3-3γ (97.3% vs. 83.8%, *p* = 0.047) and ezrin (100% vs. 86.5%, *p* = 0.021). Filamin A expression was identified less frequently in RTI-positive patients (37.8% vs. 64.9%, *p* = 0.020) similar to PARK7 expression (54.1% vs. 75.7%, *p* = 0.034). Proteins, whose expression differed significantly between RTI-positive and -negative patients in univariate analysis (Table 3) were included in multivariate analysis. Three of them were significantly associated with the risk of RTI in multivariate analysis controlling for clinical confounders: RTI was 3.5 times more likely in patients with positive 14-3-3γ expression (95% CI 1.39–28.737, *p* = 0.008), while filamin A expression lowered the risk by 0.2 times (95% CI 0.059–0.778, *p* = 0.019) and PARK7 expression by 0.3 times (95% CI 0.118–0.831, *p* = 0.020; Table 4). As expected, higher TNM stage significantly increased the risk of RTI positivity. In contrast, and surprisingly, a larger tumour diameter lowered the risk (Table 4). The model was able to correctly classify 80.6% of RTI-positive and 66.7% of -negative patients, with an overall success rate of 73.4%.

Representative photomicrographs of IHC staining and expression intensity for each examined protein in RTI-positive and -negative CS I seminoma patients are shown in Figure 2 and Appendix A, respectively. Interestingly, (i) while ezrin displayed cytoplasmatic staining in RTI-negative CS I seminomas, it stained -positive tumours in membranes, (ii) filamin A expression level was higher in the preneoplastic structures (intratubular germ cell neoplasia; IGCN) than in normal tubules (Figure 2A) and (iii) expression of all examined proteins in IC was nearly identical in both CS I seminoma groups (Figure 2B,C).

### 3.5. Expression in TGCT Cell Lines

To address the possibility of their involvement in metastatic process in TGCTs, expression of the 14-3-3β, 14-3-3γ, caldesmon, ezrin, filamin A, PARK7, vimentin and vinculin was further examined in primary tumour- and metastasis-derived TGCT cell lines (2102EP and 1777NRpmet, respectively) at the mRNA and level. As is evident (Figure 3), expression of the *YWHAB* (*p* = 0.003), *YWHAG* (*p* = 0.034), *CALD1* (*p* < 0.001), *FLNA* (*p* < 0.001), *VIME* (*p* < 0.001) and *VINC* (*p* < 0.001) genes, encoding 14-3-3β, 14-3-3γ, caldesmon, filamin A, vimentin and vinculin respectively, was significantly increased in metastasis- vs. primary tumour-derived TGCT cell lines at the mRNA level. Expression of *EZRI* (*p* = 0.263) and *PARK7* (*p* = 0.159), coding for ezrin and PARK7, respectively, remained unchanged or slightly decreased, respectively. 

## 4. Discussion

The current approach of the International Society of Urological Pathology and the American Joint Committee on Cancer classifies RTI as pathologic stage pT1 for CS I seminoma [27,28]. RTI and tumour size are being used to justify an application of single agent chemotherapy; however, this approach is controversial. Data indicate that RTI and a primary tumour size ≥4 cm correlate independently with the presence of occult (micro)metastases at diagnosis, and with a significantly increased risk of disease recurrence [12,17]. Nevertheless, the significance of RTI still remains rather a matter of discussion, and comparative genome- and proteome-wide data on RTI-positive and -negative CS I seminomas are still lacking, but remain highly important and needed.

Herein, we identify proteins that are differentially expressed in RTI-positive vs. -negative CS I seminoma using LC–MS/MS. GO analysis showed enrichment of cell–cell adhesion, and cell and biological adhesion in RTI-positive cases. The active remodelling of cell adhesion junctions by weakening strong cadherin-based cell–cell junctions and induction of expression of molecules that convey weaker and heterotypic adhesions seems to be crucial for EMT during normal morphogenesis and the process of metastasis [29,30]. To assess the clinical validity of LC–MS/MS data, the expression level of selected EMT proteins was examined in a cohort of RTI-positive and -negative CS I seminomas using IHC. Notably, RTI-positive CS I seminomas showed decreased expression of PARK7 and filamin A, when compared with -negative cases. On the other hand, ezrin and 14-3-3γ displayed an opposite association. In multivariate analysis, association between PARK7, filamin A or 14-3-3γ expression levels and the risk of RTI positivity was statistically significant. Since PARK7 expression was higher in TC with an increased mitotic activity, it might be related to tumour cell proliferation, and thus be used to monitor this process. Furthermore, based on increase of filamin A expression in IGCN compared to benign tubules, we suggest that filamin A expression, separate from other aspects (see below), might be used to reveal the initial stages of tumourigenesis in TGCTs.

The 14-3-3 group is a family of evolutionarily conserved dimeric proteins that are capable of specific phosphoserine/threonine binding to a large number of targets. Therefore, they are required for converting many phosphorylation events into subsequent biochemical/biological outcomes. Not surprisingly, 14-3-3 proteins are involved in many different cellular processes, including mitosis, cell cycle control, DNA damage checkpoint, and apoptosis [31]. Importantly, the loss of 14-3-3γ leads to cellular transformation and tumour formation in mice [32], and inhibition of migration and invasion of glioblastoma cells [33]. In an opposite manner, overexpression/up-regulation of 14-3-3γ promotes cell migration and invasion in various cancer cell lines [34,35,36] and cancer types [37,38], where it also predicts a higher probability of metastases and is associated with worse 5-year OS and PFS rates [38]. Positive correlations between 14-3-3γ expression and RTI positivity revealed herein may represent a parallel in CS I seminomas. Our findings thus suggest that 14-3-3γ is a potential candidate biomarker and therapeutic target in RTI-positive CS I seminomas. Interestingly, both the desired lowering of 14-3-3γ expression [39] and destabilization of the dimeric state of this protein [40] was shown to be experimentally achievable. The question is whether this can also be achieved for therapeutic needs.

Noteworthy, increased expression of 14-3-3γ coincides with loss of functional p53 [41]. The function of p53 in cancer is often compromised by overexpression of MDM2 or MDMX, with both homologs being negative regulators of its intracellular levels [42]. MDM2, but not MDMX, also functions as an E3 ubiquitin ligase, mediating ubiquitination and subsequent degradation of p53, MDMX, and itself [43]. Importantly, the E3 ligase activity of MDM2 is inhibited by transforming growth factor β-activated kinase 1-binding protein 1 (TAB1), whose depletion mitigates cell death caused by CDDP. Interestingly, CDDP-resistant ovarian cancer cell lines display lower TAB1 levels compared to their -sensitive counterparts [44]. Hypothetically, RTI-positive CS I seminomas expressing increased levels of 14-3-3γ may have poorer prognosis in terms of weak response to CDDP-based chemotherapy compared to -negative cases, due to the coincidental loss of p53, which may be a consequence of action of TAB1-mediated lowered E3 ligase activity of MDM2. Hence, it would be interesting to investigate the interplay of 14-3-3γ, p53, MDM2 and TAB1 in the cellular response to CDDP in RTI-positive CS I seminomas and potentially target this loop.

Filamin A cross-links F-actin filaments into dynamic orthogonal networks and interacts with the binding proteins of diverse cellular functions that are implicated in cell growth and motility regulation. Although extensively studied, the role of filamin A in cancer remains still controversial. It seems that filamin A plays a highly complex and dual role in cancer. While there are findings showing a significant decrease in filamin A levels in tissues from invasive breast cancer (BC) compared with benign disease on the one hand [45], tumorigenic enhancing activity in melanoma, lung and hepatocellular cancers has also been reported on the other [46,47,48]. Our data displaying significantly decreased expression of filamin A in CS I seminomas with RTI parallels a situation in BC, where filamin A down-regulation stimulates cancer cell migration, invasion and metastasis [45].

A dual role of filamin A in cancer has been suggested to be a consequence of its localization in the cell. When localized to the cytoplasm, filamin A has a tumour-promoting effect by interacting with signalling molecules. However, being localized to the nucleus, it may act to suppress tumour growth and inhibit metastasis by interacting with transcription factors [49]. Assuming that RTI positivity predicts the presence of occult (micro)metastases at diagnosis, our data showing virtually no change in filamin A subcellular localization upon RTI does not support a role of intracellular localization for filamin A in metastatic spread in CS I seminomas. In contrast, in prostate cancer (PC) [50] and BC [51], a process of metastasis clearly correlates with the subcellular localization of this protein. In PC, filamin A is mostly nuclear, whereas in metastatic tissue, it is mostly cytoplasmic—indicating that metastasis correlates with cytoplasmic localization of filamin A that induces cell invasion [50].

Although *FLNA* mutations in humans have contributed to our understanding of filamin A functions, knowledge of testicular functions is still very limited [52,53]. To address the role of filamin A in normal and cancerous male germ cells, in vitro and in vivo model systems knocked down for this protein have been used [54,55]. These studies have pointed to multiple functions of filamin A in normal testis and testicular cancer development, invasion and metastatic spread. In filamin A-deficient TCam-2 seminoma cells, enhanced transcript levels of the pluripotency factors OCT3/4, NANOG and FGFR3 compared to filamin A-proficient cells have been found, indicating that filamin A is involved in determining stemness in seminomas [55]. Similarly, RTI-positive CS I seminomas with significantly decreased filamin A levels likely express increased levels of these pluripotency factors, thereby activating shift from early germ cells to pluripotency phenotypes, a condition that presents a high risk for the development of invasiveness and metastasis [56]. If this assumption is true, filamin A may critically be involved in stem cell characteristics and invasiveness/metastasis in CS I seminomas.

It has been shown that reduction of filamin A sensitizes cells to DNA double-strand break (DSB)-inducing agents, ionizing radiation (IR) [57] and bleomycin [58]. Furthermore, it slows down the removal of IR-induced γH2AX nuclear foci, reduces RAD51 recruitment to chromatin in response to IR, and results in a two-fold reduction of DSB repair by homologous recombination (HR) [57], as a result of its interaction with BRCA2, a critical mediator factor in HR [58]. Based on these facts and the data of the present study, we propose that CS I seminoma patients are not managed through strategies with curative chemotherapy or radiotherapy, which are only to be applied in the case of relapse—even though this option is highly favoured by the European guidelines [59]; instead, they should be managed via a risk-adapted strategy with adjuvant DSB-based therapy in the presence of RTI. This implicates a potential of filamin A to serve as a marker for prognosis of CS I seminoma patients that are managed by DSB-based therapy.

What is the molecular basis of decreased expression of filamin A in RTI-positive CS I seminoma? First, low filamin A levels may be caused by its proteasomal degradation promoted by SQSTM1/p62 (sequestosome 1) [60]. During this process, tripartite motif containing 44 (TRIM44) protein deubiquitinates SQSTM1/p62, which leads to its oligomerization. Oligomerization prevents SQSTM1/p62 localization to the nucleus and increased cytoplasmic retention of this protein by TRIM44 prevents the degradation of filamin A [61]. Second, decreased expression of filamin A may be caused by transcriptional silencing of the *FLNA* gene through methylation, similarly to ovarian cancer, where relapse after chemotherapy is accompanied by hypermethylation of CpG islets in the promoter region of the *FLNA* gene [62]. Whatever the mechanism of decreased expression of filamin A, it increases the DNA repair capacity of the cell—a condition that worsens the responsiveness of CS I seminoma patients to DSB-based therapy. Therefore, both the epigenetic status of the *FLNA* promoter and expression of the TRIM44-SQSTM1/p62-filamin A protein loop represent potential targets for therapy-resistant CS I seminomas.

The PARK7 protein is highly conserved in a variety of mammalian tissues, and mutations in the *PARK7* gene have been found to be associated with many human diseases. PARK7 is a ubiquitous protein with multiple roles in various biological processes, including cellular transformation, signal transduction, antioxidative stress response, autophagy, apoptosis, and transcriptional regulation. Although it was originally identified as an oncogene product, the role of PARK7 in cancer is largely unknown and remains to be elucidated. Elevated PARK7 expression has been found in a variety of tumours and correlates with survival of TCs. Notably, PARK7 is secreted by TCs into the bloodstream in many cancers and can be detected in the sera of cancer patients (reviewed in [63]); its serum levels therefore correlate with disease progression and were proposed to serve as a potential prognostic biomarker in cancer. Importantly, increased bloodstream levels of PARK7 occur almost exclusively in metastatic cancer patients. On the other hand, lower levels of retained PARK7 are observed in these patients [64,65,66]. We observed lower levels of PARK7 in TCs of RTI-positive vs. -negative CS I seminoma patients. These levels likely reflect the levels of retained PARK7, suggesting that decreased levels of this protein in CS I seminoma patients may predict their poor prognosis, a suggestion in line with the assumption that RTI positivity correlates with the presence of occult (micro)metastases at diagnosis and with a significantly increased risk of disease recurrence.

PARK7 is known to protect cells against oxidative stress damage by improving mitochondrial complex I activity and, subsequently, by inhibiting mitochondria-derived reactive oxygen species production. To mediate this, PARK7 must re-localize to mitochondria [67], a process that is hypoxia-dependent [68]. Interestingly, CDDP treatment efficacy in TGCT cell lines is significantly decreased under hypoxic conditions [69] and expression of CA IX (a hypoxia marker) inversely correlates with PFS and disease recurrence in TGCT patients [70,71]. Hence, it seems that worse prognosis of RTI-positive CS I seminoma patients, manifested as a weak response to CDDP-based chemotherapy, could be associated, at least in part, with low levels of mitochondrially localized PARK7, which may be a result of overall low levels of retained PARK7.

The clinicopathological features and data on the PARK7 protein expression level from the TCGA database were used to confirm the IHC data and to review the prognostic power of PARK7 in CS I seminomas. Indeed, CS I seminoma patients from the TCGA database with PARK7 expression levels lower than the mean displayed worse OS compared to patients with higher PARK7 expression. In line with protein expression data, lower than the mean gene expression levels of *PARK7* are also associated with worse OS compared with patients with higher *PARK7* gene expression. In both cases, however, the observed association did not reach statistical significance. Therefore, more data are required to address the potential role of PARK7 in RTI-positive phenotype and prognosis of CS I seminoma patients.

We are fully aware of the discrepancy between the LC–MS/MS and IHC data, which can generate a kind of reservation about our results. However, there was substantial difference in the biological material analysed by these methods (see Materials and Methods section), and this fact represents plausible explanation for the observed discrepancy. While IHC experiments determined the expression level of selected proteins thorough the whole cell, LC–MS/MS analysis was performed on the cytoplasmic fraction only. As discussed above, the RTI process leads to changes in the subcellular localization of certain proteins, and therefore our data must be viewed in the context of these circumstances. Nevertheless, we believe that the present study brings new and valuable insights into the molecular mechanisms of RTI and potential therapeutic targets in RTI-positive CS I seminomas. However, further studies are unquestionably required to elaborate this issue in more detail and to confirm the clinical applicability of the proposed biomarkers. Logically, one of the next main goals would be evaluation of the predictive value of the proposed proteins in terms of disease recurrence in CS I seminoma patients being managed through surveillance (watchful waiting). Such experiments are currently ongoing in our laboratory.

## 5. Conclusions

A better understanding of surrounding tissue invasion and tumour dissemination is necessary for improvements in metastatic patient management. In this study, a significant difference in protein expression levels was associated with RTI in CS I seminoma. It seems that deregulated expression of filamin A, PARK7 and 14-3-3γ in RTI-positive CS I seminoma might be implicated in the pathogenesis and progression of this disease and may help to identify patients with poor prognosis. Our results support an employment of cell adhesion remodelling and the inevitable role of ECM crosslinking in RTI. However, further studies are required to confirm and extend our results, and to clarify the underlying mechanism(s). Moreover, validation of the presented results on a larger cohort of CS I seminoma patients with subsequent controlled long-term follow-up might essentially address doubts on the presence of occult (micro)metastases at diagnosis, improve prognostic stratification and bring further improvement in the management of refractory or relapsed patients. Proper implementation of the present data into prospective clinical trials might bring benefits to the individual patients involved and help to improve the therapeutic management of CS I seminoma patients in oncological centres.

## Figures and Tables

**Figure 1 cancers-13-05573-f001:**
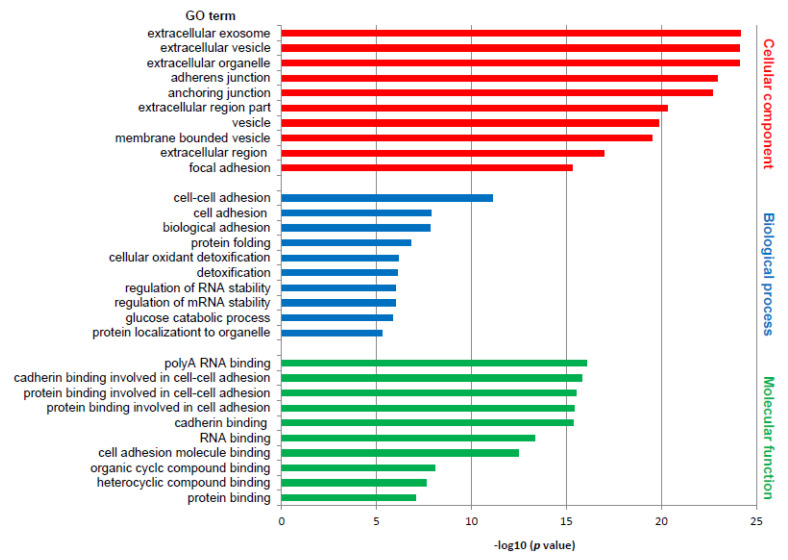
GO enrichment analysis of differentially expressed proteins in RTI-positive compared to -negative CS I seminomas. The 10 most significantly (*p* ˂ 0.05) enriched GO terms in molecular function (green), biological process (blue), and cellular component (red) are presented. All adjusted *p* values of the GO terms were −log10 transformed.

**Figure 2 cancers-13-05573-f002:**
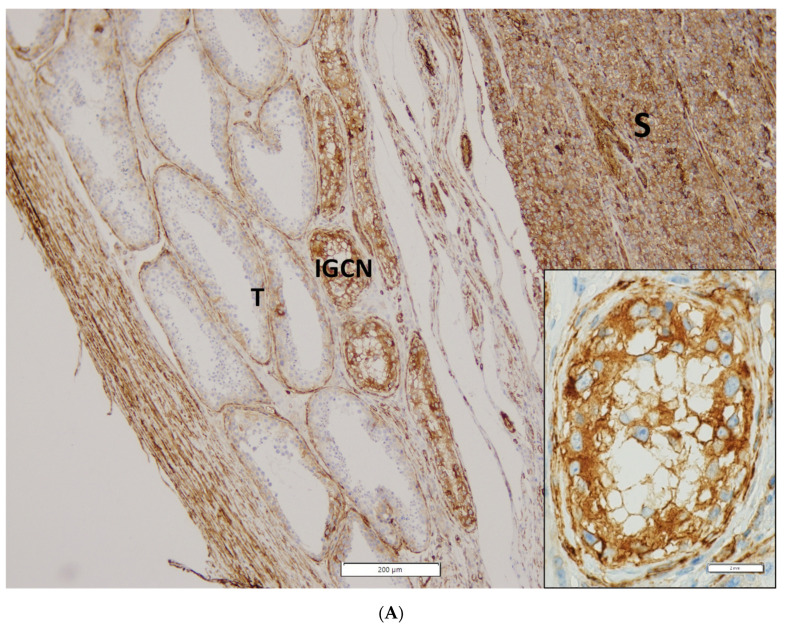
IHC of expression of 14-3-3γ, ezrin, filamin A, PARK7, vimentin and vinculin in CS I seminoma patients. Filamin A staining in seminoma (S), IGCN and benign tubules (T). Original magnification 10×. Insert: detail of filamin A positivity in IGCN, original magnification 60× (**A**). Representative photomicrographs of IHC staining of 14-3-3γ, ezrin, filamin A, PARK7, vimentin and vinculin in RTI-positive (**B**) and -negative (**C**) CS I seminoma patients. Original magnification 40×. Scale bar shown for 14-3-3γ applies to all proteins examined.

**Figure 3 cancers-13-05573-f003:**
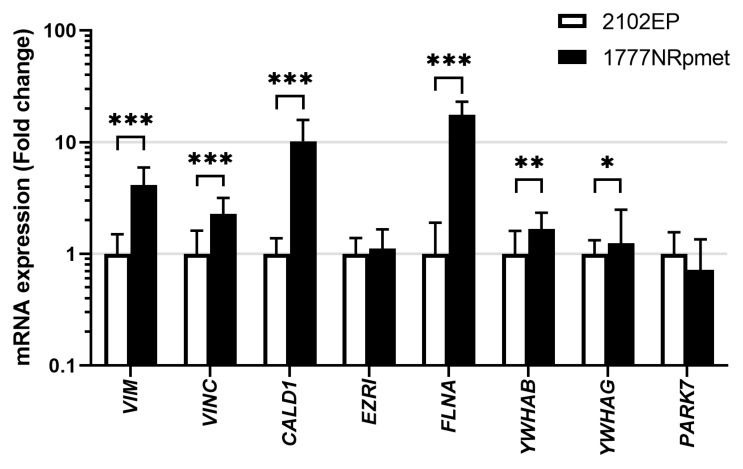
Expression of 14-3-3β, 14-3-3γ, caldesmon, ezrin, filamin A, PARK7, vimentin and vinculin at the mRNA level in primary tumour- and metastasis-derived TGCT cell lines. Expression of the 14-3-3β, 14-3-3γ, caldesmon, ezrin, filamin A, PARK7, vimentin and vinculin mRNAs in 2102EP and 1777NRpmet TGCT cell lines. Data are the means of three independent experiments. Error bars represent upper and lower limit of the expression. * *p* < 0.05; ** *p* < 0.01 and *** *p* < 0.001.

**Table 1 cancers-13-05573-t001:** Prognostic value of selected proteins (human protein database).

Protein Name	TCGA (Cancer Tissue)	Expression in Testicular Cancer	Prognostic Value
14-3-3β	All	High/Medium	Liver cancer Endometrial cancer Lung cancer Head and neck cancer Breast cancer Renal cancer
14-3-3γ	All	High/Medium	Renal cancer Cervical cancer Lung cancer Pancreatic cancer
Caldesmon	All	Weak/Negative	Renal cancer Melanoma
Ezrin	All	High/Medium	Pancreatic cancer Renal cancer Urothelial cancer Colorectal cancer Renal cancer
Filamin A	All	High/Medium	
PARK7	All	High/Medium	NA
Vimentin	All	High/Medium	Endometrial cancer Renal cancer
Vinculin	All	Medium/Weak	Pancreatic cancer

TCGA: The Cancer Genome Atlas; NA: not applicable.

**Table 2 cancers-13-05573-t002:** Clinicopathological features of RTI-positive and -negative CS I seminoma patients used in IHC studies.

Variable	RTI-Positive	%	RTI-Negative	%
All patients	37	100	37	100
TNM staging system				
pT1pNx	12	32.4	19	51.4
pT2pNx	24	64.9	18	48.6
pT3pNx	0	0	0	0
pT4pNx	1	2.7	0	0
Tumour diameter				
<4 cm	18	58.1	16	48.5
≥4 cm	13	41.9	17	51.5
Therapy modality				
Carboplatin	18	48.6	14	37.8
Radiation	14	37.8	13	35.1
Surveillance	4	10.8	8	21.6
None/No evidence	1	2.7	2	5.4
Therapy response				
Favourable (no disease progression)	35	94.6	35	94.6
Unfavourable	1	2.7	0	0
Unknown	1	2.7	2	5.4

IHC: imunohistochemistry; CS I: clinical stage I; RTI: rete testis invasive.

**Table 3 cancers-13-05573-t003:** Protein categorization during statistical analysis of IHC data.

Variable (Positivity Localization—Grade)	RTI-Positive	%	RTI-Negative	%	*p* Value
14-3-3γ (TC positivity)					**0.047**
0	1	2.7	6	16.2	
2	36	97.3	31	83.8	
Ezrin (IC positivity)					**0.021**
0	0	0	5	13.5	
2	37	100	32	86.5	
Filamin A (TC-M/C-G)					**0.020**
0	23	62.2	13	35.1	
1	14	37.8	24	64.9	
PARK7 (TC-C-G)					**0.034**
0	5	13.5	0	0	
1	12	32.4	9	24.3	
2	20	54.1	28	75.7	
Vimentin (TC positivity)					0.314
0	37	100	36	97.3	
1	0	0	1	2.7	
Vinculin (IC-G)					0.152
0	37	100	35	94.6	
1	0	0	2	5.4	

IC: intratumoural/stromal immune mononuclear cells; IC-G: intratumoural/stromal immune mononuclear cells—expression intensity grade; IHC: imunohistochemistry; RTI: rete testis invasive; TC: tumour cells; TC-C-G, tumour cells—expression intensity grade in cytoplasm; TC-M/C-G: tumour cells—expression intensity grade in membrane and/or cytoplasm. Boldface *p* value denotes statistical significance <0.05.

**Table 4 cancers-13-05573-t004:** Binary logistic regression for the relationship between the analyzed protein expression and clinicopathological characteristics with RTI.

Variable	OR	95% CI	*p* Value
Tumour diameter	0.222	0.054–0.916	**0.037**
TNM stage	5.027	1.307–19.343	**0.019**
14-3-3γ	3.488	1.392–8.737	**0.008**
Filamin A	0.215	0.059–0.778	**0.019**
PARK7	0.313	0.118–0.831	**0.020**

RTI: rete testis invasion; OR: odds ratio; CI: confidence interval; −2 Log likelihood = 65.20; R^2^ (Cox and Snell) = 0.31; R^2^ (Nagelkerke) = 0.41. Boldface *p* value denotes statistical significance <0.05.

## Data Availability

Rough data available on request.

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
