# Peer review of "Screening for the Key Proteins Associated with Rete Testis Invasion in Clinical Stage I Seminoma via Label-Free Quantitative Mass Spectrometry"

_cancers, 2021, doi:10.3390/cancers13215573_

Round 1

Reviewer 1 Report

In their manuscript Borszéková Pulzová et al. use a proteomic approach to investigate rete testis invasion (RTI)-associated proteins comparing RTI-positive versus RTI-negative clinical stage I (CS I) Seminomas. Of the differentially expressed proteins, 14-3-3γ, ezrin, filamin A, Parkinsonism-associated deglycase 7 (PARK7), vimentin and vinculin, were further analysed in CS I Seminoma patient cohort. All these proteins but 14-3-3γ result upregulated in RTI-positive using proteomic analysis (LC-MS/MS). Validation of mass spectrometry data by immunohistochemistry (IHC) reveal an opposite behaviour, PARK7 and filamin A expression were lower in RTI-positive samples, while 14-3-3γ expression was higher. Finally, the authors examined the expression of these proteins in the primary tumour- and metastasis-derived testicular germ cell tumour (TGCT) cell lines. In metastasis-derived cell line, filamin A and 14-3-3γ result significantly upregulated, according to proteomic analysis and IHC analysis respectively, while PARK7 result downregulated, according to IHC analysis. The authors conclude that de-regulated expression of filamin A, PARK7 and 14-3-3γ in RTI-positive CS I seminoma might be implicated in the pathogenesis and progression of this disease.

Despite of some typing errors the text is quite well written and the authors' hypothesis is clear. The set of experiments chosen to uncover key proteins associated with RTI useful in the Clinical management of CSI Seminoma is logical and well chosen. Unfortunately, as the authors assume in the discussion session, the experiments reveal a discrepancy between the LC-MS/MS, IHC and in vitro data (TGCT cell lines), which generate a reservation about the usefulness of the results. Moreover, the authors claim that there was substantial difference in biological material analysed by these methods trying to explain the observed discrepancy. This is true, but the authors have to standardize the results in order to clarify the possible existence and the relative behavior of potential novel biomarkers of RTI invasion, making the paper of interest for publication.

Major revision:

  • LC-MS/MS analysis: the authors have to repeat the LC-MS/MS experiment using total cell lysates and not cytoplasmic ones. Indeed, as the authors report in the discussion for filamin A protein, some proteins could have different role as a consequence of its localization in the cell. Moreover, cytoplasmic level do not necessary reflect total protein level and probably for this reason the authors observe discrepancy in the results. Importantly, the authors analyzed only six fresh tumor tissues (3 with RTI-positive and 3 with –negative CSI seminoma). Their quantitative proteomic results must be improved by increasing the number of samples using both fresh cancer tissue obtained during orchiectomy and paraffin-embedded orchiectomy specimen (for proteomic analysis from paraffin-embedded tissues there are different kits commercially available). These set of experiment can help to uniform the author’s data.

  • The Western Blot in figure S1 lack the housekeeping protein for standardization. Additionally, the data must be shown as ratio of the protein of interest to the housekeeping. Add the antibody used for the
    housekeeping in materials and methods section.

Minor revision:

  • In lane 50 (abstract) the authors assume: “To assess the role of 14-3-3γ, ezrin….”. Please, change with “to correlate the expression of…” because to assess the role of a protein the authors need to modulate a specific protein through overexpression or RNA-interference and revert the observed phenotype.
  • In Materials and Methods: Add a paragraph of TGCT cell lines information (source, growth medium, colture condition…)
  • In lane 339: Remove “protein” because The Human Protein Atlas database analyze mRNA levels from RNA-Seq (FPKM) and not protein levels.
  • In lane 372: Add (Table 4) at the end of bracket.
  • In lane 374-375: Where are these data “80.6% of RTI-positive and 66.7% of -negative Patients” in Table 4?
  • The discussion is written in a very confusing manner, it is long and sometimes off topic. Revise it.
  • In lane 497: correct “testes” with testis.
  • In lane 515: correct “Analogically” with “Similarly”.
  • In lane 609: correct “inevitable” with “necessary”.

Author Response

Dear Reviewer,

we highly appreciate your valuable comments. We did our best to address them. Hope, our reply (attached document) will fully satisfy you.

Best regards,

Miroslav Chovanec

Reviewer # 1:

In their manuscript Borszéková Pulzová et al. use a proteomic approach to investigate rete testis invasion (RTI)-associated proteins comparing RTI-positive versus RTI-negative clinical stage I (CS I) Seminomas. Of the differentially expressed proteins, 14-3-3γ, ezrin, filamin A, Parkinsonism-associated deglycase 7 (PARK7), vimentin and vinculin, were further analysed in CS I Seminoma patient cohort. All these proteins but 14-3-3γ result upregulated in RTI-positive using proteomic analysis (LC-MS/MS). Validation of mass spectrometry data by immunohistochemistry (IHC) reveal an opposite behaviour, PARK7 and filamin A expression were lower in RTI-positive samples, while 14-3-3γ expression was higher. Finally, the authors examined the expression of these proteins in the primary tumour- and metastasis-derived testicular germ cell tumour (TGCT) cell lines. In metastasis-derived cell line, filamin A and 14-3-3γ result significantly upregulated, according to proteomic analysis and IHC analysis respectively, while PARK7 result downregulated, according to IHC analysis. The authors conclude that de-regulated expression of filamin A, PARK7 and 14-3-3γ in RTI-positive CS I seminoma might be implicated in the pathogenesis and progression of this disease.

Despite of some typing errors the text is quite well written and the authors' hypothesis is clear. The set of experiments chosen to uncover key proteins associated with RTI useful in the Clinical management of CSI Seminoma is logical and well chosen. Unfortunately, as the authors assume in the discussion session, the experiments reveal a discrepancy between the LC-MS/MS, IHC and in vitro data (TGCT cell lines), which generate a reservation about the usefulness of the results. Moreover, the authors claim that there was substantial difference in biological material analysed by these methods trying to explain the observed discrepancy. This is true, but the authors have to standardize the results in order to clarify the possible existence and the relative behavior of potential novel biomarkers of RTI invasion, making the paper of interest for publication.

Major revision:

LC-MS/MS analysis: the authors have to repeat the LC-MS/MS experiment using total cell lysates and not cytoplasmic ones. Indeed, as the authors report in the discussion for filamin A protein, some proteins could have different role as a consequence of its localization in the cell. Moreover, cytoplasmic level does not necessary reflect total protein level and probably for this reason the authors observe discrepancy in the results. Importantly, the authors analyzed only six fresh tumor tissues (3 with RTI-positive and 3 with –negative CSI seminoma). Their quantitative proteomic results must be improved by increasing the number of samples using both fresh cancer tissue obtained during orchiectomy and paraffin-embedded orchiectomy specimen (for proteomic analysis from paraffin-embedded tissues there are different kits commercially available). These set of experiment can help to uniform the author’s data.

Answer:

Authors thank for this valuable comment. LC-MS/MS experiment was performed to have a gross idea about changes in the protein expression. Although the whole cell protein profile would be better to look into, we faced the masking effect in LC-MS and loss in resolution (identification) of cytoplasmic proteins. That is why the enrichment of protiens from cytoplasmic compartment was performed. 

Regarding addition of new samples – unfortunately, it is hard to get samples with similar patient background as the samples were collected a few years ago and processed fresh. We believe that including results from paraffin-embedded tissues will distort the confidence of the experiment even more.

The Western Blot in figure S1 lack the housekeeping protein for standardization. Additionally, the data must be shown as ratio of the protein of interest to the housekeeping. Add the antibody used for the housekeeping in materials and methods section.

Answer:  

As clearly stated in the manuscript, intensities of specific bands obtained by chemiluminescene were normalized to the entire UV intensity of the corresponding sample representing the total protein content. Therefore, no housekeepe/s was/were used throughout these experiments. Quantification was as follows (ezrin is shown as an example):

Please see figure in attached file.

2102EP

2102EP

2102EP

1777NRpmet

1777NRpmet

1777NRpmet

Exp. # 1

Exp. # 2

Exp. # 3

Exp. # 1

Exp. # 2

Exp. # 3

Technical replicate # 1

Total protein

301

247

331

281

303

315

ezrin

3.000

5.220

3.090

5.180

4.280

2.690

ratio

0.009967

0.021137

0.009335

0.018434

0.014125

0.007270

Technical replicate # 2

Total protein

303

200

291

249

204

303

ezrin

0.243

0.612

0.410

0.896

0.987

0.458

ratio

0.000802

0.003060

0.001409

0.003598

0.004838

0.001512

Technical replicate # 3

Total protein

177

121

336

263

160

292

ezrin

1.080

1.950

1.080

2.280

1.860

0.929

ratio

0.006102

0.016116

0.003214

0.008669

0.011625

0.003182

Three biological and three technical replicates were performed to examine expression of each protein in both TGCT cell lines. Based on the obtained ratios, fold change (FC) of protein expression was calculated and the most representative images (the ones that the most closely mirror FC between the two TGCT cell lines) were selected and are presented in Supplementary Figure 1. For this reason, we do not indeed think that it is necessary to present all the Western blots.

Minor revision:

  1. In lane 50 (abstract) the authors assume: “To assess the role of 14-3-3γ, ezrin….”. Please, change with “to correlate the expression of…” because to assess the role of a protein the authors need to modulate a specific protein through overexpression or RNA-interference and revert the observed phenotype.

Answer:

We have re-written the related sentence that reads now as follows:

In addition, the expression of 14-3-3γ, ezrin, filamin A, PARK7, vimentin and vinculin, as well as of caldesmon and 14-3-3β, was correlated in the primary tumour- and metastasis-derived testicular germ cell tumour (TGCT) cell lines.

In Materials and Methods: Add a paragraph of TGCT cell lines information (source, growth medium, culture condition…)

Answer:

The following text has been added to Materials and Methods:

Primary tumour- and metastasis-derived TGCT cell lines (2102EP and 1777NRpmet, respectively) were kindly provided Dr. Thomas Mueller (University Clinic for Internal Medicine IV, Hematology/Oncology, Medical Faculty of Martin Luther University Halle-Wittenberg, Halle, Germany). Histologically, 210EP is embryonal carcinoma and 1777NRpmet is differentiated embryonal carcinoma with immature teratoma [24, 25]. Both cell lines were grown in RPMI-1640 medium supplemented with 10% fetal bovine serum, penicillin (100 units/ml) and streptomycin (10 μl/ml). Cell lines were cultivated at 37°C in 5% CO2 atmosphere [26].

Accordingly, the following references have been added to Reference list:

  1. Bronson, D.L.; Vessella, R.L.; Fraley, E.E. Differentiation potential of human embryonal carcinoma cell lines. Cell. Differ. 1984, 15, 129-132.
  2. Andrews, P.W.; Bronson, D.L.; Benham, F.; Strickland, S.; Knowles, B.B. A comparative study of eight cell lines derived from human testicular teratocarcinoma. Int. J. Cancer 1980, 26, 269-280.
  3. Roška, J.; Wachsmannová, L.; Hurbanová, L.; Šestáková, Z.; Mueller, T.; Jurkovičová, D.; Chovanec, M. Differential gene expression in cisplatin-resistant and -sensitve testicular germ cell tumor cell lines. Oncotarget 2020, 11, 4735-4753.

In lane 339: Remove “protein” because The Human Protein Atlas database analyze mRNA levels from RNA-Seq (FPKM) and not protein levels.

Answer:

Deleted, as suggested. New sentence now reads as follows:

The Human Protein Atlas database was queried to compare our results with the known expression levels associated with cancer.

In lane 372: Add (Table 4) at the end of bracket.

Answer:

Added, as suggested. The corresponding text looks now like this:

Three of them were significantly associated with the risk of RTI in multivariate analysis controlling for clinical confounders: RTI was 3.5 times more likely in patients with positive 14-3-3γ expression (95% CI 1.392-8.737, p = 0.008), while filamin A expression lowered the risk 0.2 times (95% CI 0.059-0.778, p = 0.019) and PARK7 expression 0.3 times (95% CI 0.118-0.831, p = 0.020) (Table 4).

In lane 374-375: Where are these data “80.6% of RTI-positive and 66.7% of -negative Patients” in Table 4?

Answer:

We apologize for a mistake. You are absolutely right; these data do not appear in Table 4. Therefore, we deleted parenthesis with Table 4 at this place. Instead, we put it at the end of previous sentence. The crital two sentences read now as follows:

In contrast, and surprisingly, a larger tumour diameter lowered the risk (Table 4). The model was able to correctly classify 80.6% of RTI-positive and 66.7% of -negative patients, with an overall success rate of 73.4%.

The discussion is written in a very confusing manner, it is long and sometimes off topic. Revise it.

Answer:

Discussion has been shortened. Off topic part has been removed.

In lane 497: correct “testes” with testis.

Answer:

Done. New sentence reads now as follows:

Interestingly, filamin A has been shown to be down-regulated in the testis from rodents fed a high fat diet as compared to rodents fed a standard chow, providing an evidence for the role of diet-induced obesity in specific biological pathways and protein network in the testis.

In lane 515: correct “Analogically” with “Similarly”.

Answer:

Done. New sentence reads now:

Similarly, RTI-positive CS I seminomas with significantly decreased filamin A levels likely express increased levels of these pluripotency factors, thereby activating shift from early germ cells to pluripotency phenotype, a condition that presents a high risk for the development of invasiveness and metastasis [57].

In lane 609: correct “inevitable” with “necessary”.

Answer:

Changed, as suggested. New sentence reads now as follows:

A better understanding of surrounding tissue invasion and tumour dissemination is necessary for improvement in metastatic patient management.

Reviewer 2 Report

Article

Screening for the Key Proteins Associated with Rete Testis Invasion in Clinical Stage I Seminoma via Label-Free Quantitative Mass Spectrometry

Lucia Borszéková Pulzová, Jan Roška, Michal Kalman, Ján Kliment, Pavol Slávik, Božena Smolková, Eduard Goffa, Dana Jurkovičová, Ľudovít Kulcsár, Katarína Lešková, Peter Bujdák, Michal Mego, Mangesh R. Bhide, Lukáš Plank and Miroslav Chovanec*

In this article Pulzovà and coworkers tried to find reliable markers for risk of relapse in CS I seminoma patients. To achieve this purpose, they compared the proteomic profiles of RTI positive and negative CS I seminomas. This analysis revealed potential candidates (e.g Filamin A, PARK7 and 14-3-3y) that were validated examining their expression both at mRNA and protein levels. To assess significance of these proteins in metastatic process they finally evaluate selected protein expression in primary tumor- and metastasis-derived TGCT cell lines.

This is a potentially interesting study, conceptually relevant and original even if performed on a very small cohort (3 patients/group).

Please find below my suggestions:

Major Issue:

Figure 2 contains low qualities images taken at a low magnification; is very difficult to follow what the authors are referring to in their data (e.g the different subcellular localization between RTI + and RTI -). I strongly suggest to add some inset or to show high-magnificated pictures.

The research is largely sound and the manuscript is well written, but I think it is incomplete since actually the results are correlative rather than causative. That is of course a great start to provide hints as to a potential role of proposed proteins as clinical biomarkers, but it should be complemented at least with some functional assays to evaluate the migration capacity in TGCT cells (e.g Wound healing assay, transwell migration assay) or to evaluate the effect of overexpression/silencing of proposed candidates on EMT signalling pathways

The 14-3-3y high protein expression in RTI positive patients is unappreciable from showed IHC pictures (Fig.2B-C). Maybe a more representative picture should be shown.

In Fig.3 mRNA and protein expression of Caldesmon showed an opposite trend. This discrepancy should be at least commented in the discussion section.

Minor Issue:

Information on used cell lines and relative culture conditions are missing and should be included in Material and Methods section.

Figure 2 magnification seems to be different among pictures even if author state that the original magnification is 40X. Please check

Figure 3 statistical significance should be checked and revised since SD are very high for some candidates (e.g WHAG mRNA, 14-3-3b Protein) and it is difficult to believe they are **

Lines 497-506: the paragraph seems to be out off topic

Author Response

Dear Reviewer,

we highly appreciate your valuable comments. We did our best to address them. Hope, our reply (attached document) will fully satisfy you.

Best regards,

Miroslav Chovanec

Reviewer # 2:

In this article Pulzovà and coworkers tried to find reliable markers for risk of relapse in CS I seminoma patients. To achieve this purpose, they compared the proteomic profiles of RTI positive and negative CS I seminomas. This analysis revealed potential candidates (e.g Filamin A, PARK7 and 14-3-3y) that were validated examining their expression both at mRNA and protein levels. To assess significance of these proteins in metastatic process they finally evaluate selected protein expression in primary tumor- and metastasis-derived TGCT cell lines.

This is a potentially interesting study, conceptually relevant and original even if performed on a very small cohort (3 patients/group).

Please find below my suggestions:

Major Issue:

Figure 2 contains low qualities images taken at a low magnification; is very difficult to follow what the authors are referring to in their data (e.g the different subcellular localization between RTI + and RTI -). I strongly suggest to add some inset or to show high-magnificated pictures.

Answer:

As requested, Figure 2 was renewed accordingly to your comments. Images for ezrin and 14-3-3γ (Figure 2B and 2C) were replaced with new ones of much higher quality, so that the readership can easily follow what we are referring in the text. Also, Figure 2A has been modified, based on your suggestion – it contains insert of high magnificancy.

The research is largely sound and the manuscript is well written, but I think it is incomplete since actually the results are correlative rather than causative. That is of course a great start to provide hints as to a potential role of proposed proteins as clinical biomarkers, but it should be complemented at least with some functional assays to evaluate the migration capacity in TGCT cells (e.g Wound healing assay, transwell migration assay) or to evaluate the effect of overexpression/silencing of proposed candidates on EMT signalling pathways

Answer:

We are really happy that the reviewer appreciates our work and sees it as potentially interesting from a clinical point of view. This is exactly how we perceive our work. Therefore, and of course, we keep research on mechanisms of RTI in CS I seminoma patients ongoing in the laboratory. At this point, we indeed think that amount of work in the submitted manuscript is just enough to publish initial paper and further experimental data addressing the role of studied proteins in process of RTI in deeper detail, as well as their clinical applicability, will be included in the follow-up papers.

The 14-3-3y high protein expression in RTI positive patients is unappreciable from showed IHC pictures (Fig.2B-C). Maybe a more representative picture should be shown.

Answer:

As mentioned above, new images for 14-3-3γ now appear in the revised version of our manuscript. Hopefully, they now clearly demonstrate differential expression of this protein in RTI-positive and -negative CS I seminomas.

In Fig.3 showed an opposite trend. This discrepancy should be at least commented in the discussion section.

Answer:

We agree with you that one can implicitly assume that changes in mRNA expression have biological meaning, mediated by corresponding changes in the protein levels. However, studies into mRNA-protein correspondence have shown notoriously poor correlation between the mRNA and protein expression levels for many cellular components. Honestly, we have no satisfactory explanation for an opposite trend regarding mRNA and protein expression of caldesmon. Plausible explanation would lay in alternative splicing of the CALD1 gene resulting in multiple transcript variants encoding distinct protein isoforms. Consequently, our detection methods for the mRNA and protein level may not be equally specific for particular transcript variants and isoforms of caldesmon. Observed discrepancy is not commented in the manuscript because: (i) we have no satisfactory explanation for it, as mentioned above, and (ii) we discuss only those proteins which displayed significantly changed levels in RTI-positive compared -negative CS I seminoma examined by IHC.

Minor Issue:

Information on used cell lines and relative culture conditions are missing and should be included in Material and Methods section.

Answer:

The following text has been added to Materials and Methods:

Primary tumour- and metastasis-derived TGCT cell lines (2102EP and 1777NRpmet, respectively) were kindly provided Dr. Thomas Mueller (University Clinic for Internal Medicine IV, Hematology/Oncology, Medical Faculty of Martin Luther University Halle-Wittenberg, Halle, Germany). Histologically, 210EP is embryonal carcinoma and 1777NRpmet is differentiated embryonal carcinoma with immature teratoma [24, 25]. Both cell lines were grown in RPMI-1640 medium supplemented with 10% fetal bovine serum, penicillin (100 units/ml) and streptomycin (10 μl/ml). Cell lines were cultivated at 37°C in 5% CO2 atmosphere [26].

Accordingly, the following references have been added to Reference list:

  1. Bronson, D.L.; Vessella, R.L.; Fraley, E.E. Differentiation potential of human embryonal carcinoma cell lines. Cell. Differ. 1984, 15, 129-132.
  2. Andrews, P.W.; Bronson, D.L.; Benham, F.; Strickland, S.; Knowles, B.B. A comparative study of eight cell lines derived from human testicular teratocarcinoma. Int. J. Cancer 1980, 26, 269-280.
  3. Roška, J.; Wachsmannová, L.; Hurbanová, L.; Šestáková, Z.; Mueller, T.; Jurkovičová, D.; Chovanec, M. Differential gene expression in cisplatin-resistant and -sensitve testicular germ cell tumor cell lines. Oncotarget 2020, 11, 4735-4753.

Figure 2 magnification seems to be different among pictures even if author state that the original magnification is 40X. Please check

Answer:

According to your suggestion, magnifications in Figure 2 have been re-checked and are now quoted correctly. Some images have been replaced, as mentioned above.

Figure 3 statistical significance should be checked and revised since SD are very high for some candidates (e.g WHAG mRNA, 14-3-3b protein) and it is difficult to believe they are **

Answer:

You are absolutely right. We re-checked the significancy and found the mistakes in case of YWHAG mRNA and 14-3-3β and PARK7 protein expressions. We deeply apologize for this. New versions of Figure 3 and 4 with corrected significancies have been included in the revised version of our mauscript. In addition, legend to Figure 3 and 4 contains now one additional sentence explaining that error bars represent upper and lower limit of the expression. This type of information in extended form also appears in section Statistical analysis. Please, see revived version of our manuscript.

Statistical analysis of mRNA expression:

Gene

p value

VIME

Two-tailed P-value = 0.0000000125; N + EV - passed

VINC

Two-tailed P-value = 0.0000245; N + EV - passed

CALD1

Two-tailed P-value = 0,000000000138; N + EV -passed

ERZI

Two-tailed P-value = 0.263; N + EV - passed

FLNA

P = <0.001; N-passed, EV – failed (Mann-Whitney)

YWHAB

Two-tailed P-value = 0.00269; N + EV -passed

YWHAG

P = 0.034; N – failed (Mann-Whitney)

PARK7

Two-tailed P-value = 0.159; N + EV - passed

Statistical analysis of protein exression:

Protein

p value

Vimentin

P = 0.007; N-passed, EV- failed (Mann-Whitney)

Vinculin

P = 0.016; N-passed, EV - failed (Mann-Whitney)

Caldesmon

Two-tailed P-value = 0.0175; N+ EV -passed

Ezrin

Two-tailed P-value = 0.902; N + EV - passed

Filamin A

Two-tailed P-value = 0.218; N + EV - passed

14-3-3β

P = 0.596; N-failed (Mann-Whitney)

14-3-3γ

P = 0.007; N-passed, EV - failed (Mann-Whitney)

PARK7

P = 0.007; N-passed, EV – failed (Mann-Whitney)

N= normality test (Shapiro-Wilk); EV= equality variance test

Lines 497-506: the paragraph seems to be out off topic

Answer:

The paragraph has been deleted, as proposed.

Round 2

Reviewer 1 Report

 Dear Editor,

I have carefully read the revised manuscript and the authors' responses to my revision requests. I believe that now the manuscript has been sufficiently improved to warrant publication in Cancers.

Best regards

Reviewer 2 Report

The authors have satisfied my requests by making the required changes if possible. For this reason I believe that the article can be accepted in the present form.